# Effect of Iron Supplementation on the Modulation of Iron Metabolism, Muscle Damage Biomarkers and Cortisol in Professional Cyclists

**DOI:** 10.3390/nu11030500

**Published:** 2019-02-27

**Authors:** Alfredo Córdova, Juan Mielgo-Ayuso, Cesar I. Fernandez-Lazaro, Alberto Caballero-García, Enrique Roche, Diego Fernández-Lázaro

**Affiliations:** 1Department of Biochemistry and Physiology, Faculty of Physical Therapy, University of Valladolid, Campus de Soria, 42003 Soria, Spain; a.cordova@bio.uva.es; 2Department of Biochemistry and Histology, Faculty of Physical Therapy, University of Valladolid, Campus de Soria, 42003 Soria, Spain; fernandezlazaro@usal.es (C.I.F.-L.); diego.fernandez.lazaro@uva.es (D.F.-L.); 3Department of Preventive Medicine and Public Health, School of Medicine, University of Navarra, 31008 Pamplona, Spain; 4Department of Anatomy and Radiology, Faculty of Physical Therapy, University of Valladolid, Campus de Soria, 42003 Soria, Spain; albcab@ah.uva.es; 5Department of Applied Biology-Nutrition and Institute of Bioengineering, Alicante Institute for Health and Biomedical Research (ISABIAL-FISABIO Foundation), 03202 Elche, Alicante, Spain; eroche@umh.es; 6CIBERobn (Fisiopatología de la Obesidad y la Nutrición CB12/03/30038) Instituto de Salud Carlos III, 28029 Madrid, Spain

**Keywords:** recovery, iron supplementation, exercise, cyclist, muscular biomarkers, performance

## Abstract

Background: The intense efforts made during 3-week stage races may reduce iron metabolism and hematological parameters. These efforts may increase the levels of circulating muscle damage markers and some hormones. All of these physiological changes may have negative consequences not only for the performance of athletes but also for their health. The main aim of this study was to evaluate the effects of supplementation with 80 mg/day of iron on haematological parameters, serum cortisol and biochemical muscle indicators on elite male cyclists during the 3-week stage race the *Vuelta a España*. Our secondary aim was to examine whether the hematological profile is associated with muscular damage parameters and cortisol. Methods: Eighteen elite male cyclists from two teams were randomly assigned to one of two groups: (1) control group (CG, *n* = 9; age: 26.1 ± 4.6 years; maximum oxygen uptake per kg: 78.0 ± 5.4 mL/kg/min) or (2) group treated with 80 mg/day iron (800 mg of iron protein succinylate, ITG, *n* = 9; age: 25.7 ± 6.4 years; maximum oxygen uptake per kg: 77.6 ± 6.5 mL/kg/min). The cyclists were subjected to blood tests one week before the start of the race (T1) and after 4 weeks of treatment, coinciding with the end of the competition (T2). Iron metabolism parameters, muscle damage indicators and serum cortisol were assessed. Repeated-measures ANOVA with group as a factor (GC and ITG) were used to examine the differences between groups throughout the study (time × group) after iron supplementation treatment. Results: Significant differences were observed between groups throughout the study in the group-by-time interaction and changes in serum iron (GC: −8.93 ± 10.35% vs. ITG: 0.60 ± 8.64%; *p* = 0.018), ferritin (GC: −13.88 ± 23.53% vs. ITG: 91.08 ± 118.30%; *p* = 0.004), haemoglobin (GC: 10.00 ± 3.32% vs. ITG: 13.04 ± 5.64%; *p* < 0.001), haematocrit (GC: −1.17 ± 3.78% vs. ITG: 7.32 ± 3.92%; *p* < 0.001) and cortisol (GC: 24.74 ± 25.84% vs. ITG: –13.54 ± 13.61%; *p* = 0.005). However, no significant group-by-time interaction was observed for the circulating muscle biomarkers. Additionally, significant negative correlations of serum iron, haemoglobin and haematocrit with muscle circulating biomarkers and cortisol (*p* < 0.05) were observed. Conclusions: Oral iron supplementation with 80 mg/day iron (800 mg of iron protein succinylate) effectively prevented a decline in haematological parameters (serum iron, ferritin, haemoglobin and haematocrit) and maintained optimal levels of recovery in elite cyclists during the *Vuelta a España*. Moreover, the hematological values were shown to have relationship with muscular recovery parameters.

## 1. Introduction

Iron is an essential micronutrient that plays a significant role in physical performance, particularly in demanding endurance sports, such as cycling [1,2,3]. Of special interest are long stage races in which iron metabolism, hematological parameters, muscle enzymes and hormones undergo changes for which impacts on cyclists’ health have not been assessed [1,4]. The intense and sustained efforts made during 3-week stage races may reduce blood hemoglobin (Hb), ferritin (FER) and serum iron (sFe) concentrations in endurance cyclists, resulting in marginal or deficient iron states, one of the main causes of fatigue in this discipline [1,5,6]. In addition, sustained and intense exercise may increase the levels of circulating muscle damage markers, such as serum creatine kinase (CK), lactate dehydrogenase (LDH) and myoglobin (Mb) as well as hormones, such as cortisol, which is involved in the activation of catabolic processes and anti-anabolic actions related to protein turnover [7]. These parameters are indicative of increased muscle damage and oxidative stress, which negatively affect athletes’ performance and increase the duration required for optimal recovery [4,8,9,10]. As a result, athletes not only may reduce their performance, but also may put their health at risk.

Oral iron supplementation is an effective nutritional strategy that may be considered for individuals with suboptimal FER levels and for preventing iron deficiency [11]. In addition to an adequate iron intake from the regular diet, iron supplementation (~80 mg/day) can be considered to prevent decreases in iron levels and related parameters in athletes undergoing a high level of effort [12]. This supplementation has shown improvements in FER and the hematological profile during training periods in different sports [11,12,13,14]. It is well-known that endurance stage races, such as the *Tour de France* [5] or the *Giro d’Italia* [1] reduce hematological parameters. However, to our knowledge, iron supplementation has not been tested during a 3-week cycling race.

Recent studies have determined a relationship between skeletal muscle and body iron metabolism [15]. In fact, increased iron availability triggered by low hepcidin levels may provide iron levels that contribute to muscle growth [15]. However, no authors have demonstrated iron supplementation to be a successful nutritional strategy to reduce muscle damage and stress. We hypothesize that oral iron supplementation will be a useful ergogenic aid to improve hematological profiles and reduce muscle damage and stress in non-anemic athletes.

The aim of this study was to evaluate the effects of 80 mg/day of oral iron supplementation (800 mg of iron protein succinylate) on the hematological profile (sFe, FER, transferrin saturation index (TSI), hematocrit (Htc), and Hb), muscle damage intracellular enzymes (CK, LDH, Mb) and cortisol in elite male cyclists over a time-span of 4 weeks during the *Vuelta a España*. Our secondary aim was to examine the relationship between the hematological profile and biochemical muscular damage parameters and cortisol. We hypothesize that iron supplementation will prevent any decline in the hematological profile and subsequently, decrease muscle damage.

## 2. Materials and Methods

### 2.1. Sample

Eighteen professional male cyclists from two different elite teams participated in the study. Both teams had similar objectives and competed in the 3-week cycling stage race known as the *Vuelta a España*, which covers around 3300 km. Medical examinations were conducted by the Union Cyclist International (UCI) to ensure participants were free of disease prior to the beginning of the study. To the authors’ knowledge, no banned drugs or medications were taken by the riders. None of the athletes tested positive for the routine doping tests performed before and during the race according to the World Anti-Doping Agency (WADA). 

All participants had analogous dietetic and training programs to prepare the race and were supervised by a team physician in collaboration with one of the authors of the study. Physiotherapeutic and nutritional post-exercise recovery strategies were also pretty similar. All participants were informed of the experimental procedures, risks and benefits and voluntarily signed a written consent. The study was designed according to the Declaration of Helsinki for experiments with human beings and was approved by the local Ethics Committee of University of Valladolid.

### 2.2. Experimental Protocol and Assessment Plan

A randomized and non-controlled by placebo design was applied to analyse the effects of oral iron supplementation on the hematological profile, biomarkers of muscular damage and cortisol levels. 

Participants were assigned to groups using a stratified block design. An independent statistician generated the following random allocation sequence: (a) a control group (CG, *n* = 9, age: 26.1 ± 4.6 years; maximum oxygen uptake per kg: 78.0 ± 5.4 mL/kg/min) that did not receive oral iron, (b) group treated with 80 mg/day iron (ITG, *n* = 9, age: 25.7 ± 6.4 years; maximum oxygen uptake per kg: 77.6 ± 6.5 mL/kg/min).

The cyclists were tested at two specific points during the study: at baseline one week before the start of the race (T1) for diet/supplementation homogenization and at the beginning of the last stage (T2) (3rd week of race). Therefore, the total time between T1 and T2 was 4 weeks, including the first week being for homogenization and the remaining 3 weeks for the race. The ITG received 80 mg/day oral iron in the form of 800 mg of iron protein succinylate (Ferplex 40^®^, Italfarmaco S.A., Madrid, Spain). Iron supplementation was provided in doses of 40 mg twice per day according to the laboratory´s instructions during breakfast and dinner in the 4-week period. 

Protein succinylate was chosen in order to avoid problems related to tolerance, gastrointestinal discomfort, bioavailability and food/supplement compatibility [16]. During the competition, the research team ensured that constant circadian rhythms were maintained in terms of nutrition, hydration, timing of food intake and sleep, which is a normal practice in cycling competitions. 

### 2.3. Dietary Assessment

A professional dietitian strictly recorded the daily food and fluid intake of the cyclists during the study. The EasyDiet^©^ package, which is available online (https://www.easydiet.es/), was used to calculate the nutrient composition and energy intake from the foods and drinks consumed by the cyclists. This software package was developed by the Spanish Centre for Higher Studies in Nutrition and Dietetics (CESNID) and is based on the Spanish tables of food composition [17]. In addition, all participants received multivitamin pills that included folic acid (5 mg/day), vitamin C (1 g/day), B12 vitamin (1000 µg/day), branched amino acids and glutamine.

### 2.4. Body Composition and Anthropometric Measures

Anthropometric measurements were taken by an ISAK (International Society for Advancement in Kinanthropometry) level 3 anthropometric device, following the standard procedures in T1. The height and body weight technical error of measurement was less than 0.02%, and it was less than 2.6% in the case of skinfolds. The height (cm) was measured with a SECA^©^ measuring rod (Barcelona, Spain), with a precision of 1 mm (range: 130–210 cm), and body weight (kg) was assessed by a SECA^©^ scale, with a precision of 0.1 kg (range: 2–130 kg). All skinfolds (triceps, biceps, abdominal, supraspinal, subscapular, chest, front thigh and medial calf) were taken using a Harpenden skinfold caliper (CMS instruments, London, UK), with a precision of 0.2 mm. The sum of 6 skinfolds (triceps, abdominal, supraspinal, subscapular, front thigh and medial calf) was calculated as a reference for fat content.

### 2.5. Blood Collection and Analysis 

Blood extraction and transportation were performed according the UCI and WADA guidelines (www.ama-wada.org). All samples were collected under basal conditions after a 10–12 h overnight fast at T1 and T2. The blood samples (15 mL) were obtained at 8:30 a.m. from the antecubital vein with the subject seated in a comfortable position using Vacutainer tubes. Blood was distributed in one tube with gel and clot activator (10 mL) to obtain serum, and other EDTA tubes (3–5 mL) were used to obtain plasma. 

Immediately after filling, EDTA tubes containing blood were inverted 10 times and stored in a sealed box at 4 °C. Temperature control was assured during transportation; the specific tag Libero Ti1-Elpro (Buchs, Switzerland) was used for temperature measurement and recording. The EDTA anticoagulated blood was processed according to UCI and WADA recommendations [18]. Blood in EDTA tubes was centrifuged at 2000 rpm for 15 min. Plasma (upper layer) was extracted using a Pasteur pipette, transferred to a sterile tube and stored at −20 °C until the moment of analysis.

Red blood cells (RBC), Hb and Hct were determined in the Hematology Analyser Sysmex XE-2100 (Barcelona, Spain). 

Serum from blood samples was separated by low-speed centrifugation and analysed according to Wilkinson et al. [19]. Total serum protein was determined by a colorimetric method using an autoanalyser Hitachi 917 (Tokyo, Japan). Serum Fe and total iron-binding capacity were measured in an automated chemistry analyser Synchron CX (Beckman Coulter Diagnosis, Madrid, Spain). FER was determined in duplicate using the IRMA kit (Bio-Rad, Madrid, Spain). Transferrin (TRF) was measured by nephelometry (Boehringer–Mannheim, Barcelona, Spain). Finally, the transferrin saturation index (TSI) (%) was calculated as the ratio of sFe to the total iron binding capacity.

Serum levels of CK and LDH were measured using coupled enzyme reactions on an automatic autoanalyser Hitachi 917. Mb assessment was performed using a chemiluminescence immunoassay. Cortisol levels were measured by an enzyme-linked fluorescent assay in a multiparametric analyser Minividas^®^ (Biomerieux, Marcy l’Etoile, France) using 4 methyl umbelipherone as a substrate with stimulation at 370 nm and emission at 450 nm.

### 2.6. Statistical Data Analyses

Statistical analyses were performed using the using SPSS Statistics software (SPSS: An IBM Company, version 24.0, IBM Corporation, Armonk, NY, USA) and Graphpad Prism (Graphpad Software version 6, San Diego, CA, USA). Data were expressed as mean ± standard (SD) deviation. Significant differences were established at *p* < 0.05. Differences between groups for hematological parameters, muscle damage and cortisol parameters were assessed by independent Student’s *t* tests after normality had been confirmed with the Shapiro–Wilk test, opting for a parametric analysis. Likewise, parametric paired *t*-tests were used with the blood sample parameters to identify significant differences between T1 and T2 in each group independently. Additionally, a two-way repeated measure analysis of variance (ANOVA) was carried out using the Greenhouse–Geisser test to confirm the existence of an interaction effect (time × group) between CG and ITG for hematological parameters, muscle damage and cortisol parameters throughout the study (T1 and T2). Effect sizes among participants were calculated using partial eta square (*η*^2^*p*). Since this measure is likely to overestimate effect sizes, values were interpreted in accordance with Ferguson [20], whereby no effect is indicated by 0 ≤ *η*^2^*p* < 0.05; a minimum effect is indicated if 0.05 ≤ *η*^2^*p* < 0.26; a moderate effect is indicated if 0.26 ≤ *η*^2^*p* < 0.64; and a strong effect is indicated if *η*^2^*p* ≥ 0.64. Lastly, correlations between hematological parameters and muscular biomarkers were examined using Pearson´s correlation coefficient in order to estimate the association between two variables. 

## 3. Results

Table 1 depicts the anthropometric and physiological characteristics of the participants at T1. No significant differences between groups were found (*p* > 0.05).

The average daily energy and micronutrient intake in CG and ITG during the 4 weeks of study can be observed in Table 2. No significant differences between groups for total caloric intake and micronutrients, including iron, were found (*p* > 0.05).

Table 3 shows the mean ± SD of certain hematological parameters directly or indirectly related to iron homeostasis. These include TRF, TSI, total serum proteins and RBC in CG and ITG at T1 and T2. Statistically significant differences in the group-by-time interaction of TSI (*p* = 0.025; *η*^2^*p* = 0.248), TP (*p* = 0.017; *η*^2^*p* = 0.290), and RBC (*p* < 0.001; *η*^2^*p* = 0.815) between the two groups through the study were observed in favour of ITG. Along this line, ITG presented significant increases between T1 and T2 (*p* < 0.05) in TSI (T1: 35.5 ± 10.0 vs. T2: 31.1 ± 12.9%), TP (T1: 6.9 ± 0.3 vs. T2: 7.1 ± 0.3 g/L) and RBC (T1: 4.8 ± 0.3 vs. T2: 5.3 ± 0.2 × 10^6^/µL). However, CG did not present significant differences between T1 and T2 for any hematological parameters (*p* > 0.05).

Figure 1 displays the mean ± SD of other hematological parameters that are related to iron homeostasis. These include sFe, FER, Hb and Hct in CG and ITG at T1 and T2. The results show statistical differences in the group-by-time interaction for sFe (*p* = 0.018; *η*^2^*p* = 0.271), FER (*p* = 0.004; *η*^2^*p* = 0.395), Hb (*p* < 0.001; *η*^2^*p* = 0.723) and Hct (*p* < 0.001; *η*^2^*p* = 0.586) in favor of ITG. As a result, significant increases were observed in ITG with respect to CG at T2 (sFe: CG: 98.9 ± 15.8 vs. ITG: 116.7 ± 14.4 µg/dL; FER: CG: 104.7 ± 41.2 vs. ITG: 231.0 ± 62.0 ng/mL; Hb: CG: 15.0 ± 1.2 vs. ITG: 16.6 ± 0.5 g/L; Hct: CG: 44.1 ± 3.0 vs. ITG: 49.7 ± 2.0 %; all *p* < 0.05). The absence of significant differences at T1 reinforces the homogeneity of the participants at the beginning of intervention T2 (sFe: CG: 108.6 ± 39.6 vs. ITG: 116.0 ± 21.6 µg/dL; FER: CG: 128.3 ± 55.7 vs. ITG: 153.3 ± 68.8 ng/mL; Hb: CG: 15.1 ± 0.8 vs. ITG: 14.7 ± 0.9 g/L; Hct: CG: 44.6 ± 2.5 vs. ITG: 46.3 ± 2.6%; all *p* > 0.05). On the other hand, while CG only presented significant differences in FER between T1 and T2 (*p* < 0.05), ITG presented significant increase (*p* < 0.05) in FER, Hb and Hct.

Figure 2 depicts the mean ± SD of circulating muscle damage markers (CK, LDH and Mb) and cortisol, which is considered to be a stress marker, during T1 and T2 in CG and ITG. Regarding muscle markers, there were no significant differences in the group-by-time interaction between the two groups through the study (T1 vs. T2). However, when comparing within groups, CK decreased in the ITG at the end of the intervention period (T1: 306.7 ± 56.12 UI/L vs. T2: 260.8 ± 66.3 UI/L; *p* < 0.05). LDH (T1: 378.3 ± 112.2 vs. T2: 296.6 ± 89.6 UI/L), and to a lesser degree Mb (T1: 23.3 ± 6.3 vs. T2: 24.2 ± 5.8 ng/mL) presented a tendency to decrease in ITG at T2, although the differences were not significant (*p* > 0.05). No changes were noticed for CK, LDH and Mb in the CG when comparing T1 vs. T2.

Regarding cortisol, there were significant differences in the group-by-time interaction between the two groups throughout the study (T1 vs. T2) (*p* = 0.005; *η*^2^*p* = 0.495). Concretely, this parameter significantly increased only in the CG at the end of the intervention period (T1: 15.4 ± 3.2 vs. T2: 19.3 ± 6.1 mg/dL). However, in ITG, there was a tendency for circulating cortisol to decrease, although the difference between T1 and T2 was not significant (T1: 16.2 ± 5.9 mg/dL vs. T2: 15.1 ± 2.3 mg/dL; *p* > 0.05).

Correlations between hematological parameters related to iron homeostasis and biochemical muscle markers are shown in Table 4. High negative correlations were observed for sFe with LDH (r = −0.944; *p* < 0.001), Mb (r = −0.924; *p* < 0.001), and CK (r = −0.915; *p* < 0.001), and there was a moderate negative correlation with C (r = −0.596; *p* = 0.011). In the same way, there were higher negative correlations between Hb and all muscular markers: LDH (r = −0.986; *p* < 0.001), Mb (r = −0.994; *p* < 0.001), CK (r = −0.996; *p* < 0.001). Moreover, Hb showed a higher negative correlation with hormone C (r = −0.910; *p* < 0.001). Finally, FER also showed moderate-high negative correlations with LDH (r = −0.643; *p* = 0.003), Mb (r = −0.0643; *p* = 0.003) and CK (r = −0.701; *p* = 0.001) and a high negative correlation with C (r = −0.962; *p* > 0.001). 

Lastly, Figure 3 displays changes at the end of the study (T2–T1) in terms of the percentage of all parameters analysed in Figure 1 and Figure 2. A positive value indicates that the value of the parameter at the end (T2) is higher than at the beginning (T1), and vice versa. Regarding hematological parameters, there were significantly greater increases (*p* < 0.05) in favour of ITG compared to CG for sFe (CG: −8.93 ± 10.35% vs. ITG: 0.60 ± 8.64%), Hb (CG: −0.41 ± 3.32% vs. ITG: 130.4 ± 5.64%), Hct (CG: −1.17 ± 3.78 vs. ITG: 7.32 ± 3.92%) and FER (CG: −13.88 ± 23.53% vs. ITG: 91.08 ± 118.3%). On the contrary, the percentage changes for muscle damage proteins such as CK (CG: 28.27 ± 100.25% vs. ITG: −53.53 ± 29.18%) and Mb (CG: 3.86 ± 7.94% vs. ITG: −4.15 ± 8.15%) and cortisol (CG: 24.74 ± 25.84% vs. ITG: −13.54 ± 13.61%) were significantly greater increases (*p* < 0.05) for CG compared with ITG.

## 4. Discussion

To our knowledge, this is the first study to describe the effects of iron supplementation on the modulation of iron-related metabolic parameters, circulating muscle biomarkers and cortisol in a population of elite cyclists during a large competitive race such as the *Vuelta a España*. The main findings of the study were that supplementation with 80 mg/day iron significantly increases sFe, FER, Hb and Hct without changes in muscle biomarkers. Cortisol tended to decrease in the ITG, while it significantly increased in the CG at the end of the race, indicating a higher level of effort-induced stress. A significant negative correlation was observed for LDH, Mb and CK with sFe and Hb.

During the 3-week long stage competition, the metabolic homeostasis of the athletes suffered a large amount of stress. Particularly, hematological parameters were modified due to the effort, and consequently, by the adaptations required for the highly demanding performances [5,21]. Along this line, Mørkeberg et al. [5] and Corsetti et al. [1] reported significant decreases in Hb and Hct during the *Tour de France* and *Giro d´Italia,* respectively, but no data regarding the use of supplementation to correct these problems was reported. In our study, we did not find any significant differences in sFe, Hb and Hct in the CG between T1 and T2; however, we observed a decreasing tendency for all haematological parameters. To avoid this situation, a recent review suggests that iron supplementation may be useful to prevent iron deficiency in ultra-endurance athletes [22]. Likewise, iron supplementation could prevent fatigue in subjects with iron depletion without anaemia [23], even when the athletes take adequate amounts of iron in the diet [12]. We supplemented cyclists with 40 mg twice per day because previous studies [23] have demonstrated that doses higher than 60 mg of iron increase hepcidin levels and consequently, decrease iron absorption. We also decided to use iron proteinsuccinylate (Ferplex 40^®^) because it enhances iron bioavailability by up to 35% and reduces the gastrointestinal problems and discomfort that are generally associated with intake of other iron supplements [24,25]. Altogether, the results seem to support the recommendations made by other authors, indicating that supplementation with 80 mg/day iron (800 mg of iron protein succinylate) could not only prevent a decline but could also improve hematological parameters, including both Htc and Hb, during 3-week long stage race such as the *Vuelta a España*. 

Cumulative fatigue as a consequence of bouts of sustained and intense exercise compromises muscle integrity, favouring the release of muscle-damage proteins, such as CK, Mb and LDH, to the bloodstream. These proteins are regularly used as markers of muscle stress due to vigorously intense exercise [4]. Similarly, although professional cyclists generally have higher LDH and CK values at rest compared to sedentary individuals [26], these parameters were shown to further increase after 2 weeks of intense training [27]. This observation indicates that muscle integrity, and thereby function, could be strongly compromised under prolonged, highly intensive exercise, such as in large cycling competitions. Therefore, to maintain the cyclists’ health, it is important to preserve muscle integrity as much as possible. It was observed that muscle integrity seemed to be preserved under more optimal conditions in the ITG. However, the CG showed an increasing tendency for muscular damage parameters to be present, consistent with previous studies [4,8,9,10]. In fact, circulating CK levels in the ITG were significantly lower at the end of the race (T2) than at the beginning (T1). The other muscle markers (LDH and Mb) also presented this tendency, but the differences between T1 and T2 were not significant. This could be explained by the different half-lives that released muscle proteins have in circulation: 1–1.5 days for CK and 6–7 days for LDH and Mb [28]. However, the differences observed in these circulating parameters when comparing both groups (CG vs. ITG) were not significant, suggesting that iron supplementation could help to prevent muscle damage, although it is likely that other complementary strategies need to be considered to preserve muscle integrity during highly demanding efforts. Other studies have corroborated the benefits of iron supplementation. For example, Dely et al. indicated lower CK values in both adults and children after aseptic skeletal muscle trauma when they were supplemented with iron [29]. Other studies, such as that of Dallman et al., which was performed in rats, showed a normalization of Mb levels when sFe deficiency was corrected through iron supplementation [30]. Altogether, these results reinforce the idea that iron supplementation could avoid, or at least ameliorate, muscle damage. However, future studies are necessary to adapt these recovery strategies to the particular situation that occurs in demanding sport disciplines such as endurance cycling competitions.

Although the molecular mechanisms have not been addressed in the present study, it is evident that there is a correlation between adequate iron homeostasis and optimal muscle function [15]. This is manifested by the need for oxygen delivery to the working muscles as well as mitochondrial energy production through Hb and Mb. Therefore, it is evident that an optimal iron balance is instrumental for the majority of the physiological variables involved in athletic performance [31]. Moreover, iron metabolism is also instrumental in post-exercise recovery periods. In this context, a more adequate recovery may lead to a greater training volume and skeletal muscle adaptations, and iron supplementation strategies could also accelerate the recovery time between sessions [12]. Consistently, a previous study described decreases in sFe, Hb and Hct levels together with a significant increase in the circulating muscle biomarkers CK and LDH in elite athletes after an ultra-marathon competition [32]. These results were very similar to the observations reported in the present study. In addition, the correlations calculated could illustrate this connection between muscle integrity and iron metabolism [15]. Therefore, iron supplementation could be a good strategy to increase iron stores and enhance endurance capacity by means of improving hematological parameters and therefore decreasing fatigue and muscle damage [15]. In any case, in order to optimize performance and recovery in disciplines that require very intensive physical effort, future research needs to decipher the molecular mechanisms involved [33].

Serum cortisol levels are considered to be an indicative factor of accumulated stress in response to intensity [8,34]. The level of response is modulated by both the relative intensity and duration of the exercise [35]—higher intensity and longer duration correspond to greater cortisol release. Moreover, it has been indicated that when the relationship training/recovery is well graduated, cortisol levels return to basal levels within 24 h post-exercise [10,34,36]. Similarly, Hough et al. observed higher cortisol values after 11 days of a highly intense cycling training period, indicating high levels of stress and inadequate recovery [37]. In the present study, we observed a significant increase in basal serum cortisol in CG, but not in ITG after 3 weeks of racing, indicating a lower level of stress and, most likely, better recovery in ITG. In another study, Clemente showed a decrease in cortisol and hematological parameters 5 days after an ultra-endurance race [35]. Therefore, low cortisol levels seem to be necessary to modulate the exercise-induced inflammatory response [38] that is instrumental for muscle repair/regeneration and optimal recovery to allow better future performances [39].

Our study has some limitations. First, we did not record the total time taken by the cyclists to complete the *Vuelta a España*. Consequently, we were not able to relate the hematological and muscle damage values to their performance. However, the particular role of each cyclist in the team during the race made it difficult to address this interesting point. In this context, a long tour involves a great effort by all individuals of the team. Individual and collective objectives can modulate the effort of each particular member to a greater or lesser extent, which is translated into different physiological responses and thereby performances.

Other limitation of the study is that we did not determine the plasma level of hepcidin. Plasma hepcidin is negatively correlated with iron bioavailability, and iron supplementation acutely increases circulating plasma hepcidin [40]. However, recently, Ihibashi et al. [41] reported that the effect of moderate iron supplementation on the hepcidin level during strenuous training remains unclear. Along this line, however, our results showed a statistical difference in the group-by-time interaction for sFe in favour of ITG, indicating that the influence of hepcidin could be low.

## 5. Conclusions

In summary, oral supplementation with 80 mg/day iron (800 mg of iron protein succinylate) over a 4 week period (week before and 3 weeks during a big tour—Vuelta a España) was shown to be sufficient to prevent a decline in hematological levels (ferritin, hemoglobin and hematocrit) and to reduce accumulated stress in response to exercise intensity (cortisol). Although 80 mg/day iron (800 mg of iron protein succinylate) over four weeks was not enough to show a higher group-by-time interaction of muscle damage biomarkers, it was observed that hematological levels were associated with muscle damage biomarkers. 

## Figures and Tables

**Figure 1 nutrients-11-00500-f001:**
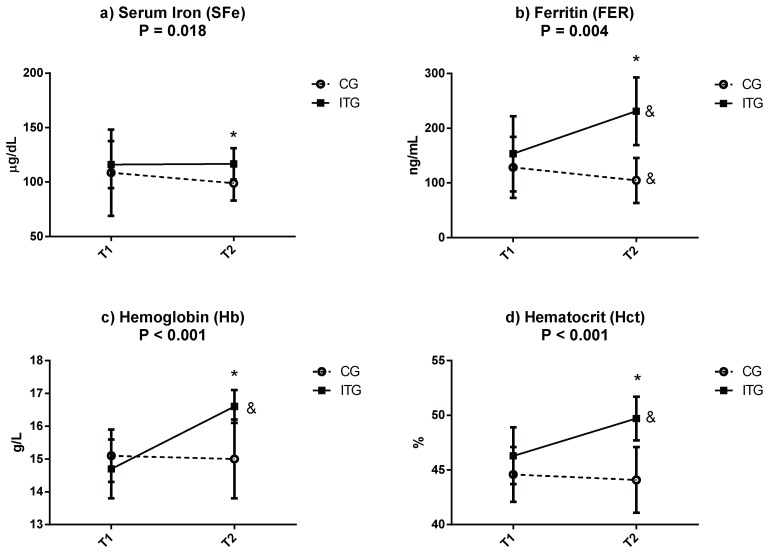
Concentrations of (**a**) serum iron (sFe), (**b**) ferritin (FER), (**c**) hemoglobin (Hb) and (**d**) hematocrit (Hct) in the control group (CG) and iron treatment group (ITG) at baseline (T1) and after 3 weeks (T2). Data are expressed as mean ± standard deviation. *p*: group-by-time interaction (*p* < 0.05, all such occurrences). Two-factor repeated-measures ANOVA. &: Significant difference between phases as determined by dependent *t*-tests (T1 vs. T2), *p* < 0.05. *: Significantly different between groups as determined by independent *t*-tests (CG vs. ITG), *p* < 0.05.

**Figure 2 nutrients-11-00500-f002:**
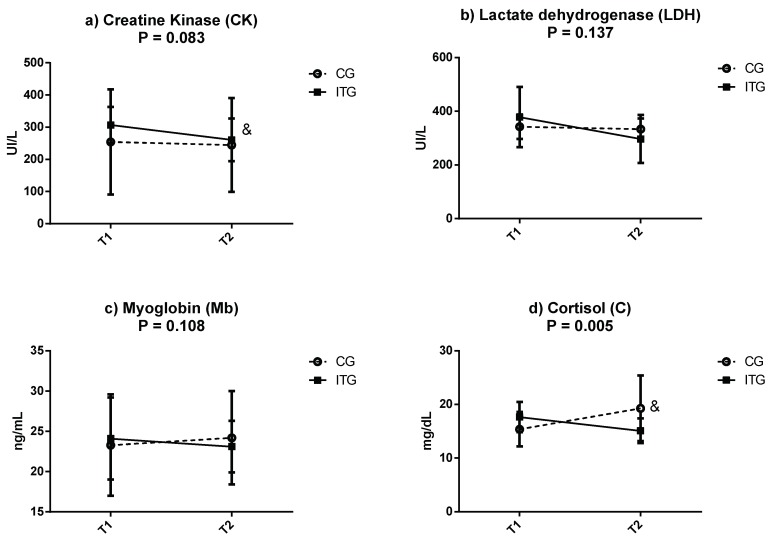
Concentrations of (**a**) creatine kinase (CK), (**b**) lactate dehydrogenase (LDH), (**c**) myoglobin (Mb) and (**d**) cortisol (C) in the control group (CG) and in the iron treated group (ITG) at the beginning (T1) and at the end of intervention (T2). Data are expressed as mean ± standard deviation. P: group-by-time interaction (*p* < 0.05, all such occurrences). Two-factor repeated-measures ANOVA. &: Significantly different between phases as determined by dependent *t*-test (T1 vs. T2), *p* < 0.05.

**Figure 3 nutrients-11-00500-f003:**
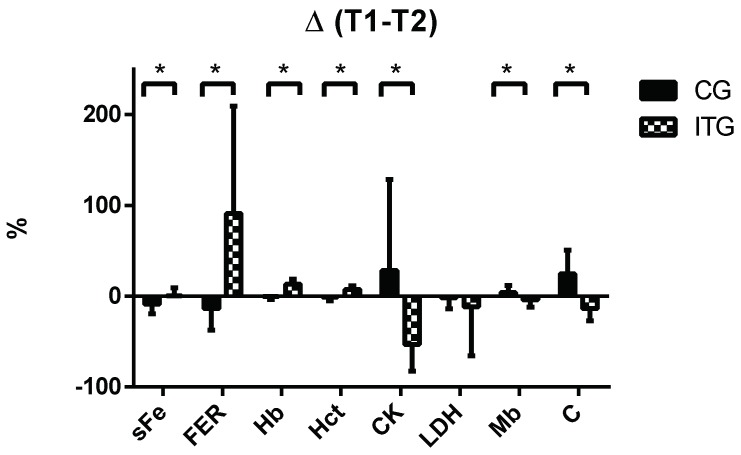
Percentages of change observed during the study in iron-related parameters, muscle damage markers and cortisol in the control group (CG) and the iron supplemented group (ITG). Data are expressed as mean ± standard deviation. sFe: Serum iron; FER: Serum ferritin; Hb: Hemoglobin; Hct: Hematocrit; CK: Creatine Kinase; LDH: Lactate dehydrogenase; Mb: Myoglobin; C: Cortisol. *: Significantly different between groups, *p* < 0.05.

**Table 1 nutrients-11-00500-t001:** Physical and anthropometric characteristics in the control group (CG) and iron treatment group (ITG) at baseline (T1).

	CG	ITG	*p*
Age (years)	26.1 ± 4.6	25.7 ± 6.4	0.880
Weight (kg)	69.3 ± 5.6	69.9 ± 7.9	0.854
Height (cm)	179.1 ± 12.8	178.2 ± 16.3	0.898
Σ 6 skinfolds (mm)	35.2 ± 7.5	35.4 ± 8.2	0.956
Maximum oxygen uptake (mL/kg/min)	78.0 ± 5.4	77.6 ± 6.5	0.889

Data are expressed as mean ± standard deviation. *p*: Differences between groups as determined by independent *t*-tests (CG vs. ITG).

**Table 2 nutrients-11-00500-t002:** Average daily energy and micronutrient intakes in the control group (CG) and iron treatment group (ITG) during the 4 weeks of study.

	GC	ITG	*p*
Energy (kcal/kg)	76.4 ± 5.8	75.4 ± 4.7	0.693
Calcium (mg)	2153 ± 222	2336 ± 257	0.126
Magnesium (mg)	1030 ± 109	1042 ± 72	0.786
Phosphorus (mg)	3928 ± 84	4032 ± 291	0.318
Iron (mg)	43.2 ± 5.6	43.7 ± 4.0	0.830
Zinc (mg)	26.5 ± 0.8	27.0 ± 2.2	0.531
Vitamin A (μg)	3512 ± 1270	3186 ± 585	0.494
Vitamin E (mg)	32.4 ± 3.5	30.6 ± 2.9	0.252
Thiamine (mg)	4.86 ± 0.20	4.64 ± 0.54	0.269
Riboflavin (mg)	5.08 ± 0.22	5.31 ± 0.54	0.254
Niacin (mg)	70.2 ± 6.8	69.1 ± 7.3	0.745
Vitamin B6 (mg)	7.18 ± 0.62	7.04 ± 0.97	0.720
Folic Acid (μg)	1177 ± 176	1175 ± 158	0.980
Vitamin B12(μg)	17.62 ± 3.89	16.61 ± 4.38	0.611
Vitamin C (mg)	650 ± 148	702 ± 158	0.482

Data expressed as mean ± standard deviation. *p*: Differences between groups as determined by independent *t*-tests (CG vs. ITG).

**Table 3 nutrients-11-00500-t003:** Hematological parameters and total protein in the control group (CG) and iron treatment group (ITG) at baseline (T1) and after 3 weeks (T2).

Group	T1	T2	*p*	*η* ^2^ *p*
**Transferrin (mg/dL)**				
CG	243.3 ± 25.4	234.4 ± 24.2	0.195	0.097
ITG	232.0 ± 18.6	242.3 ± 44.4		
**Transferrin Saturation Index (%)**				
CG	31.7 ± 9.6	29.9 ± 8.9	0.025	0.248
ITG	35.5 ± 10.0	31.1 ± 12.9 ^&^		
**Total Protein (g/dL)**				
CG	7.0 ± 0.3	7.0 ± 0.6	0.017	0.290
ITG	6.9 ± 0.3	7.1 ± 0.3 ^&^		
**Red Blood Cells (×10^6^ cells/µL)**				
CG	4.9 ± 0.4	4.9 ± 0.4	<0.001	0.815
ITG	4.8 ± 0.3	5.0 ± 0.2 ^&^		

Data are expressed as mean ± standard deviation. *p*: group-by-time interaction (*p* < 0.05, all such occurrences). Two-factor repeated-measures ANOVA. No significant differences between groups (CG vs. ITG) were found by independent *t*-tests. ^&^ Significant differences (*p* < 0.05) over time (T1 vs. T2) within the same group as determined by dependent *t*-tests.

**Table 4 nutrients-11-00500-t004:** Correlations between hematological biomarkers (serum iron (sFe), hemoglobin (Hb) and ferritin (FER)) and muscular biochemical markers (lactate dehydrogenase (LDH), myoglobin (Mb), creatine kinase (CK) and cortisol (C)).

	sFe	Hb	FER
LDH	−0.944 **	−0.986 **	−0.643 *
Mb	−0.924 **	−0.994 **	−0.643 *
CK	−0.915 **	−0.996 **	−0.701 *
C	−0.596 *	−0.910 **	−0.962 **

Data are expressed by Pearson´s coefficient. * *p* < 0.005. ** *p* < 0.001.

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
