# Peer review of "Effect of Iron Supplementation on the Modulation of Iron Metabolism, Muscle Damage Biomarkers and Cortisol in Professional Cyclists"

_nutrients, 2019, doi:10.3390/nu11030500_

Reviewer 1 Report

Reviewer Report:

Title: Effect of iron supplementation on the modulation of iron metabolism, muscle damage markers and cortisol in professional cyclists

Overview:

The article provides an insight into iron supplementation in 18 elite cyclist during a stage race. Supplementation with 80 mg/day of Iron Protein Succinylate for 4 weeks maintained iron status in the supplemented group of elite cyclist during the staged cycle race. However, the supplementation was unable to prevent the rise in circulating muscle biomarkers during the 4 weeks.
Results from this study may suggest that iron supplementation may be used to prevent the onset of fatigue associated with iron depletion in athletes that may result from continuous and strenuous endurance training. I would like to commend the authors for their efforts in this research study and value the direct sporting application this study has for elite endurance athletes. Below are some comments that the authors may consider for their final journal submission.

Revisions:

Line 77: please consider changing ‘none authors’ to ‘no authors’

Methods/Results: Are you able to provide the exercise and performance data (GPS, training load, intensity or duration) from the race as a supplementary file to the article. If so are you able to comment on any differences between the CG and the ITG in overall training intensity, load and or duration of cycling during each week and whether it could have exacerbated the muscle damage, inflammatory response and contributed to the findings of the current study.

Methods, Line 118: Are you able to provide a rational for the split dose for the iron supplementation. In a recent review on iron measurement and supplementation (Clenin 2015), a single dose in the morning every second day in athletes without iron deficiency was suggest as an optimal treatment method to ensure optimal iron absorptions without augmenting hepcidin levels. Please can you provide additional reasoning for your method of supplementation in this non-iron deficient cohort.

Methods Line 120: Iron protein succinylate appears to be less potent that iron salts. In line with the previous comment, if split doses will augment the hepcidin response and reduce absolute iron absorption, would the authors mind justifying their current protocol for supplementation.

Results: Line 202. Do the authors mean ITG presented a significant increase between T1 and T2, if so please correct to T2.

Results: Line 215-217 authors will need to reword the sentence as it currently reads as though the CG and ITG group were significantly different at T2. However your results suggest that there was no statistical significance between the groups in any parameters at T2 yet large effect sizes would favour and increase/improvements in the ITG group. Please ensure that you accurately present your results and ensure that you only state that results are significantly different if they truly are. In all other cases please ensure you let the reader know that the magnitude of difference (effect size) was greater for the ITG group. Please revise your results section and ensure that you have accurately presented all your results and discussed trends versus statistical significance correctly.

Discussion Line 292-296: please address the length and grammar of this sentence

Line 295: Please correct Htc to Hct

Limitations: authors may wish to add in a section on hepcidin and suggest that they did not measure it and hence the extent to which iron absorption may have been compromised was unable to be established.

Author Response

Dear Journal Editor and Reviewer,

We appreciate again the time you devoted to reading our manuscript and helping us to craft an improved version. We are pleased to clarify your concerns which we believe have improved the quality and applicability of your work. Please, find below our response to your observations. We have made a concerted attempt to systematically address the specific concerns raised for this revision and we have highlighted the alterations to this revision within the manuscript in green for your convenience.

Reviewer 1 Report:

Title: Effect of iron supplementation on the modulation of iron metabolism, muscle damage markers and cortisol in professional cyclists

Overview:

The article provides an insight into iron supplementation in 18 elite cyclist during a stage race. Supplementation with 80 mg/day of Iron Protein Succinylate for 4 weeks maintained iron status in the supplemented group of elite cyclist during the staged cycle race. However, the supplementation was unable to prevent the rise in circulating muscle biomarkers during the 4 weeks.

Results from this study may suggest that iron supplementation may be used to prevent the onset of fatigue associated with iron depletion in athletes that may result from continuous and strenuous endurance training. I would like to commend the authors for their efforts in this research study and value the direct sporting application this study has for elite endurance athletes. Below are some comments that the authors may consider for their final journal submission.

Revisions:

Line 77: please consider changing ‘none authors’ to ‘no authors’

Answer: Thank you for your suggestion. The authors have addressed the concern.

Methods/Results: Are you able to provide the exercise and performance data (GPS, training load, intensity or duration) from the race as a supplementary file to the article. If so are you able to comment on any differences between the CG and the ITG in overall training intensity, load and or duration of cycling during each week and whether it could have exacerbated the muscle damage, inflammatory response and contributed to the findings of the current study.

Answer: Thank you for your suggestion. In Our study, the cyclists were RAMDOMLY assigned to one of the groups (CG or ITG) so we do not see why there should be differences in overall training intensity, load and or duration of cycling during each week. Then, both groups (CG & ITG) should have similar activity.

 However, we are not able to provide any information about the performance of the cyclists. We believe thateach cyclist plays a different  role in the team depending on the team strategies and they do not have the same goals. For instance, while the leader or leaders of the team have the goal of winning the competition, the goal of their teammates is to help their leaders. In our study (18 professional male cyclists of two different elite teams), all cyclists of both groups (CG and ITG), made efforts to the maximum of their capacities during 21 consecutive days and all cyclists completed the 3500 km. Therefore, none of the cyclists has experimented muscle damage associated with a reduction in power output or cycling performance. Cyclists’ average pulse/day was 140-150 bpm after (this is the only information of exercise and performance that  the authors of the article have). We believe that both groups are comparable because ofthe reasons that we have previously described.

This was pointed out in the limitation section as follows: Our study has some limitations. First, we did not have the total time of the cyclists at the end of the Vuelta a España. Consequently, we were not able to relate the hematological and muscle damage values with their performance. (page 13 Lines 374-376).  However, the particular role of each cyclist in the team during the race made it difficult to address this interesting point. In this context, a long tour involves a great effort by all individuals of the team. Individual and collective objectives can modulate the effort of each particular member in a greater or lesser extension, which is translated to different physiological responses and thereby performances.

 Methods, Line 118: Are you able to provide a rational for the split dose for the iron supplementation. In a recent review on iron measurement and supplementation (Clenin 2015), a single dose in the morning every second day in athletes without iron deficiency was suggest as an optimal treatment method to ensure optimal iron absorptions without augmenting hepcidin levels. Please can you provide additional reasoning for your method of supplementation in this non-iron deficient cohort.

Answer: Thank you for your concern. We supplemented cyclist with 40 mg twice per day rather than a single dose because previous studies [23] have demonstrated that doses higher than 60 mg of iron increase hepcidin levels and consequently, decrease iron absorption. We also decided to use iron proteinsuccinylate (Ferplex 40®) as supplementation because it reduces the gastrointestinal problems and discomfort that it is generally associated with other iron intake supplements [24]. This is particularly important in elite cyclists that need to focus 100% on the race and cannot make a stop during the stage because of nauseas, vomiting, or diarrhea. Furthermore, iron proteinsuccinylate enhances iron bioavailability up to 35% compared to 10-15% that is absorbed from ferrous iron salt preparations or food [25]. Moreover, we strictly followed the laboratory (ITALFARMACO S.A) recommendations for the iron supplementation: “use of 2 drinkable vials per day split in two doses, preferably before meals”.

The authors of the article agree with reviewer that a single dose in the morning every second day in athletes without iron deficiency as suggested by Clein et al. (2015) is an optimal treatment. However, our study was performed in elite athletes that have a high endurance demand during 3 weeks, that means, over 3500 kms and more than 80 hours of intense exercise in just 21 days.

Rational for the dose of iron supplementation has been introduced in the manuscript as follows: “We supplemented cyclist with 40 mg twice per day because previous studies [23] have demonstrated that doses higher than 60 mg of iron increase hepcidin levels and consequently, decrease iron absorption. We also decided to use iron proteinsuccinylate (Ferplex 40®) because it enhances iron bioavailability up to 35% and reduces gastrointestinal problems and discomfort that it is generally associated with other iron intake supplements [24,25]Page 11 line 309-314

 1.         Ishibashi A, Maeda N, Kamei A, Goto K. Iron supplementation during three consecutive days of endurance training augmented hepcidin levels. Nutrients. 2017;9(8):820.

2.         Peeling P, Dawson B, Goodman C, Landers G, Wiegerinck ET, Swinkels DW, et al. Effects of exercise on hepcidin response and iron metabolism during recovery. International journal of sport nutrition and exercise metabolism. 2009;19(6):583-97.

3.         Moretti D, Goede JS, Zeder C, Jiskra M, Chatzinakou V, Tjalsma H, et al. Oral iron supplements increase hepcidin and decrease iron absorption from daily or twice-daily doses in iron-depleted young women. Blood. 2015:blood-2015-05-642223.

4.         Mielgo-Ayuso J, Zourdos M, Calleja-González J, Córdova A, Fernandez-Lázaro D, Caballero-García A. Eleven Weeks of Iron Supplementation Does Not Maintain Iron Status for an Entire Competitive Season in Elite Female Volleyball Players: A Follow-Up Study. Nutrients. 2018;10(10):1526.

5.         Roecker L, Meier-Buttermilch R, Brechtel L, Nemeth E, Ganz T. Iron-regulatory protein hepcidin is increased in female athletes after a marathon. European journal of applied physiology. 2005;95(5-6):569-71.

6.         Nielsen P, Nachtigall D. Iron supplementation in athletes. Current recommendations. Sports Med. 1998;26(4):207-16.

7.         Villanueva J, Soria M, Gonzalez-Haro C, Ezquerra L, Nieto JL, Escanero JF. Oral iron treatment has a positive effect on iron metabolism in elite soccer players. Biol Trace Elem Res. 2011;142(3):398-406.

Methods Line 120: Iron protein succinylate appears to be less potent that iron salts. In line with the previous comment, if split doses will augment the hepcidin response and reduce absolute iron absorption, would the authors mind justifying their current protocol for supplementation.

Answer: This concern has been addressed in the previous comment.

Results: Line 202. Do the authors mean ITG presented a significant increase between T1 and T2, if so please correct to T2.

Answer: The authors have now addressed this issue: “In this line, ITG presented significant increase between T1 and T2 (P < 0.05) on TSI (T1: 35.5±10.0 vs. T2: 31.1±12.9%), TP (T1: 6.9±0.3 vs. T2: 7.1±0.3 g/l) and RBC (T1: 4.8±0.3 vs T2: 5.3±0.2x106/µl).” page?? line??

Results: Line 215-217 authors will need to reword the sentence as it currently reads as though the CG and ITG group were significantly different at T2. However, your results suggest that there was no statistical significance between the groups in any parameters at T2 yet large effect sizes would favour and increase/improvements in the ITG group. Please ensure that you accurately present your results and ensure that you only state that results are significantly different if they truly are. In all other cases please ensure you let the reader know that the magnitude of difference (effect size) was greater for the ITG group. Please revise your results section and ensure that you have accurately presented all your results and discussed trends versus statistical significance correctly.

Answer: Thank you for your suggestion. The authors modified figure 1 and figure 2 to show that there were significant differences in the hematological values of T2 between CG and ITG.

Discussion Line 292-296: please address the length and grammar of this sentence

Answer: We have introduced the following changes: “Altogether, the results seem to support the recommendation made by other authors, indicating that an iron supplementation with 80 mg/day (800 mg of Iron Protein Succinylate) could not only prevent a decline, but also improve hematological parameters, including both Htc and Hb, during 3-week long stage race such as the Vuelta a España.” Page 11 line 313 and Page 12 Lines 314-316

Line 295: Please correct Htc to Hct

Answer: The authors have addressed this issue.

Limitations: authors may wish to add in a section on hepcidin and suggest that they did not measure it and hence the extent to which iron absorption may have been compromised was unable to be established.

Answer: As suggested by the reviewer, the authors have introduced the following limitation: “Other limitation of the study is that it has not determined plasma levels of hepcidin. Plasma hepcidin negatively correlates with iron bioavailability and iron supplementation acutely increases the circulating plasma hepcidin [40].” Page 13 lines 380- 383

Reviewer 2 Report

The paper from Cordova et al investigates a potentially interesting topic: to increase iron availability to improve erythroid activity and muscle stress during professional physical activity.

The paper results to me difficult to read as a whole and to interpret the results.

My main concern regards the values found in the control group (CG). The authors could not find any differences between the T1 and T2 values in 3 out of four of the haematological parameters (sFe, Hb and Hct) in the control groups (Figure 1) even if they are expected to change on the basis of the cited literature (1,5).

The same result is even more evident for the muscle damage specific markers (CK,LDH, and Mb) (Figure 2)and again this data is not in line with the literature (4, 7-9). The ITG group reaches at T2 values very close to the CG group at the same time point.

Only cortisol is normalized at T2 in ITG group but it could be due to several factors independent from muscles.

On these basis one could conclude that the oral iron supplementation in ITG causes an increase in body iron amount (increased FER, Hb and Hct) with no advantage on muscles. Not to talk to the fact that increased Hct could be very risky for the anti-doping laws.

Since no significant differences are reported on athletes performance (line 351-352), it difficult for me to extrapolate a final message.

 Abstract must be carefully reviewed and significantly modified because it seems to be confused and superficial and it does not encourage to continue reading the paper.

Line 23-25: sentence not clear and grammatically incorrected, several words repeated. Please list clearly the factors that you analysed in your work : haematological parameters, serum cortisol and biochemical muscle indicators.

Lines 36-40: all the parameters values should be transferred in the results section, where they are not reported

Line 41. “it has observed” must be corrected in “it was observed” and moved at the end of the sentence.

Introduction

Lin 61. Please add some more information on cortisol role as specific muscle stress marker.

Line 80: …..profiles as well AS muscle damage-

Results

Line 202 ……between T1 and T2

Table 3. I would align the P and h2p to the ITG values because the refers to this group specifically.

Line 212 on: it will be more understandable if you present for the mean ±SD (as you said in MM section) the significance and the h2p values for any haematological parameter you analyse taking them from the abstract section.

Author Response

Dear Journal Editor and Reviewer,

We appreciate again the time you devoted to reading our manuscript and helping us to craft an improved version. We are pleased to clarify your concerns which we believe have improved the quality and applicability of your work. Please, find below our response to your observations. We have made a concerted attempt to systematically address the specific concerns raised for this revision and we have highlighted the alterations to this revision within the manuscript in green for your convenience.

Reviewer 2 Report:

The paper from Cordova et al investigates a potentially interesting topic: to increase iron availability to improve erythroid activity and muscle stress during professional physical activity.

The paper results to me difficult to read as a whole and to interpret the results.

Answer: The authors have introduced new changes and re-wrote the results section to enhance the flow of the manuscript and facilitate the interpretation of the results.

My main concern regards the values found in the control group (CG). The authors could not find any differences between the T1 and T2 values in 3 out of four of the haematological parameters (sFe, Hb and Hct) in the control groups (Figure 1) even if they are expected to change on the basis of the cited literature (1,5).

Answer: Thank you for your comment.  We agree with the reviewer that we could not find any significant difference in the hematological values of the CG. However, the results showed a decreasing tendency over time. We have introduced a new sentence into the manuscript to address the reviewer’s concern: “In our study, we did not find any significant difference in the sFe, Hb, and Hct of the CG between T1 and T2, however, we observed a decreasing tendency for all the haematological parameters” (page 11, line 304-306)

The same result is even more evident for the muscle damage specific markers (CK, LDH, and Mb) (Figure 2) and again this data is not in line with the literature (4, 7-9). The ITG group reaches at T2 values very close to the CG group at the same time point.

Answer: Thank you for your feedback. We did not find any significant difference in any damage muscle markers in the CG but we did find an increasing tendency. The authors have introduced the following sentence in the manuscript to address reviewer’s concern:However, the CG showed an increasing tendency in muscular damage parameters consistent with previous studies [4,8-10].” (page 12, line 327-328)

Only cortisol is normalized at T2 in ITG group but it could be due to several factors independent from muscles.

Answer: Thank you for your concern. The authors have introduced a new sentence to clarify that all participants of the study had similar physiotherapeutic and nutritional post-exercise recovery strategies: “Moreover, all participants in the study had similar physiotherapeutic and nutritional post-exercise recovery strategies”. (page 3, line 102-103)

On these basis one could conclude that the oral iron supplementation in ITG causes an increase in body iron amount (increased FER, Hb and Hct) with no advantage on muscles. Not to talk to the fact that increased Hct could be very risky for the anti-doping laws.

Answer: Thank you for your observation. The authors think that due to the modest bioavailability, the increases in body iron could be modest, but sufficient to correct some parameters during the race. Cyclists were only supplemented for 4 weeks, including a first week for homogenization and the remaining 3 weeks for the race, in this short period of competition it will be unlikely that cyclists to exceed the 50% hematocrit level of the anti-doping laws. In addition, none of the cyclists of our study tested positive during the La Vuleta España, and it is well-known that cyclists generally have to take more doping test controls than any other athletes. This fact suggests that if athletes follow the guidelines described in the intervention group, they will not have any problem with doping laws.

Since no significant differences are reported on athletes performance (line 351-352), it difficult for me to extrapolate a final message.

Answer: Thank you for your concern. To improve clarity, the authors have introduced new edits in the limitations section: Our study has some limitations. First, we did not have the total time of the cyclists at the end of the Vuelta a España. Consequently, we were not able to relate the hematological and muscle damage values with their performance”. (Page 13 Lines 374-376)

Abstract must be carefully reviewed and significantly modified because it seems to be confused and superficial and it does not encourage to continue reading the paper.

Answer: Thank you for your suggestion. The authors have reviewed and improved the the abstract.

Line 23-25: sentence not clear and grammatically incorrected, several words repeated. Please list clearly the factors that you analysed in your work : haematological parameters, serum cortisol and biochemical muscle indicators.

Answer: Thank you for your suggestion. The authors have modified the manuscript and introduce the following sentence: “The main aim of the study was to evaluate the effects of 80 mg/day of iron supplementation on haematological parameters, serum cortisol and biochemical muscle indicators on elite male cyclists during the Vuelta a España.” (Page 1 Lines 24-26)

Lines 36-40: all the parameters values should be transferred in the results section, where they are not reported

Answer: Thanks for the recommendation. The authors have included the values of the parameters in the results section.

Line 41. “it has observed” must be corrected in “it was observed” and moved at the end of the sentence.

Answer: We have modified the manuscript according reviewer’s suggestion: On the other hand, a significant negative correlation of serum iron, haemoglobin and haematocrit with muscle circulating biomarkers and cortisol (P < 0.05) was observed.” (Page 1 Line 46) (Page 2 Line 47)

 Introduction

 Lin 61. Please add some more information on cortisol role as specific muscle stress marker.

Answer: Thank you for your suggestion. The authors have explained the role that cortisol plays as a specific muscle stress marker: “In addition, sustained and intense exercise may increase the levels of circulating muscle damage markers, such as serum creatine kinase (CK), lactate dehydrogenase (LDH) and myoglobin (Mb), and hormones such as cortisol which is involved in activation of catabolic processes and anti-anabolic actions related to protein turnover [7].” (Page 2 Line 61-65)

Line 80: …..profiles as well AS muscle damage-

Answer: The authors have addressed this issue.

Results

 Line 202 ……between T1 and T2

Answer: Thank you for your correction. This has been addressed.

Table 3. I would align the P and h2p to the ITG values because the refers to this group specifically.

Answer: Thanks for your comment. The authors want to clarify that P and η2p is of the group-by-time interaction. In other words, it indicates if the behavior of the different parameters was significantly different between GC and ITG throughout the study (between T1 and T2).

Line 212 on: it will be more understandable if you present for the mean ±SD (as you said in MM section) the significance and the h2p values for any haematological parameter you analyse taking them from the abstract section.

Answer: Thank you for your recommendation. The authors have followed the reviewer’s recommendations.

Reviewer 3 Report

This is a well written, clear and interesting paper. I agree with the authors about the limitations of their study due to the lack of the valuation of the correlation between ITG and performance during the race. I also understand their difficulty to address this point in that specific context. In my opinion, however, this pivotal work could offer the basis for further investigation on the role of Iron Succinylate oral supplementation during cyclist performances. 

Author Response

Reviewer 3 Report:

This is a well written, clear and interesting paper. I agree with the authors about the limitations of their study due to the lack of the valuation of the correlation between ITG and performance during the race. I also understand their difficulty to address this point in that specific context. In my opinion, however, this pivotal work could offer the basis for further investigation on the role of Iron Succinylate oral supplementation during cyclist performances.

Answer: On behalf of all co-authors, many thanks for the insightful comments and suggestions for this review.

Round  2

Reviewer 2 Report

The authors answers to my requests are adequate and sufficient.